# The *DBB* Family in *Populus trichocarpa*: Identification, Characterization, Evolution and Expression Profiles

**DOI:** 10.3390/molecules29081823

**Published:** 2024-04-17

**Authors:** Ruihua Wu, Yuxin Li, Lin Wang, Zitian Li, Runbin Wu, Kehang Xu, Yixin Liu

**Affiliations:** 1College of Biological Sciences and Technology, Beijing Forestry University, Beijing 100083, China; 13073712372@163.com (L.W.); lizitian2388@bjfu.edu.cn (Z.L.); wrb19800369792@163.com (R.W.); xukehang2000@bjfu.edu.cn (K.X.); 2Melbourne School of Design, The University of Melbourne, Parkville, VIC 3010, Australia; yuxinl19@student.unimelb.edu.au; 3College of Landscape Architecture and Art, Northwest A & F University, Yangling 712100, China

**Keywords:** DBB, *Populus trichocarpa*, phylogenetic relationships, expression patterns, stress response

## Abstract

The B-box proteins (BBXs) encode a family of zinc-finger transcription factors that regulate the plant circadian rhythm and early light morphogenesis. The double B-box (*DBB*) family is in the class of the B-box family, which contains two conserved B-box domains and lacks a CCT (CO, CO-like and TOC1) motif. In this study, the identity, classification, structures, conserved motifs, chromosomal location, *cis* elements, duplication events, and expression profiles of the *PtrDBB* genes were analyzed in the woody model plant *Populus trichocarpa*. Here, 12 *PtrDBB* genes (*PtrDBB1*–*PtrDBB12*) were identified and classified into four distinct groups, and all of them were homogeneously spread among eight out of seventeen poplar chromosomes. The collinearity analysis of the DBB family genes from *P. trichocarpa* and two other species (*Z. mays* and *A. thaliana*) indicated that segmental duplication gene pairs and high-level conservation were identified. The analysis of duplication events demonstrates an insight into the evolutionary patterns of *DBB* genes. The previously published transcriptome data showed that *PtrDBB* genes represented distinct expression patterns in various tissues at different stages. In addition, it was speculated that several *PtrDBBs* are involved in the responsive to drought stress, light/dark, and ABA and MeJA treatments, which implied that they might function in abiotic stress and phytohormone responses. In summary, our results contribute to the further understanding of the DBB family and provide a reference for potential functional studies of *PtrDBB* genes in *P. trichocarpa*.

## 1. Introduction

Transcription factors (TFs) are important regulatory proteins that act on various stages of plant growth and development and regulate gene expression under stress conditions by activating or inhibiting the transcription of target genes [1,2]. According to the data provided by PlantTFDB, the poplar genome encodes 2455 putative TFs and can be divided into 58 subfamilies. Moreover, zinc finger transcription factors are a large gene family of TFs, which can be divided into different subfamilies based on the motif structures and functional characteristics of each member [2,3]. The B-box (BBX) protein belongs to a subfamily of the zinc finger TFs, which participate in transcriptional regulation by binding to other proteins or DNA [4].

The BBX transcription factor is characterized by the existence of one or two B-box domains at the N-end of its protein and CCT domains at the C-end [5,6]. They are considered to be involved in the regulation of seedling photomorphogenesis, plant phototropism and flowering induction and affect hormone-signaling pathways [7,8]. The double B-box (DBB) transcription factor carries two B-box domains (B-box1 and B-box2) in the N-terminal region without a CCT domain. Moreover, the length of each B-box is about 40 amino acids, and 8–15 amino acids are present between the two B-box domains [5,9]. 

The Arabidopsis *DBB* gene family consists of eight members: *DBB1a* (At2G21320, *BBX18*), *DBB1b* (At4G38960, *BBX19*), *DBB2* (At4G39070, *BBX20*), *DBB3* (At1G78600, *BBX22*), *DBB4* (At4G10240, *BBX23*), *STO* (At1G06040, *BBX24*), *STH* (At2G31380, *BBX25*), and *STH2* (At1G75540, *BBX21*) [10]. Previous reports have shown that *DBB* genes control the optical signal transduction pathway during early-stage photomorphogenesis process, and the transcription levels of many genes in this *DBB* family are controlled by circadian rhythms, such as *DBB1a*, *DBB1b*, *DBB2*, *DBB3*, *STO* and *STH* [11]. The gene expression of these genes after light induction was different; among them, *DBB1a* (*BBX18*), *DBB3* (*BBX22*), *STO* (*BBX24*) and *STH* (*BBX25*) are induced by light and *DBB1a* is induced by light to the largest extent, whereas *DBB2* (*BBX20*) is inhibited by light, and *DBB1b* (*BBX19*) is basically not regulated by light [12,13]. However, these genes are involved in regulating the expression of *HY5*, *COP1* and *CHS* in the light-signaling pathway [14,15] as well as the expression of key genes in the circadian clock signaling pathway, such as *CCA1*, *LHY*, *ELF3* and *TOC1*, and they played an important role in light-mediated plant growth [16]. *STO* (*BBX24*) and *STH* (*BBX25*) regulate a plant’s salt tolerance and play a negative role in the phytochrome and blue light signal transduction pathways [17]. *DDB3* (*BBX22*) and *STH2* (*BBX21*) interact with two key regulators of light signaling, *HY5* and *COP1*, and they serve as an active regulator of hypocotyl elongation, early chloroplast formation and anthocyanin accumulation in the early stage of plant growth [10,18]. Additionally, 12 *DBB* genes have been identified in maize, and an increasing body of results imply that plant DBB proteins might function in the light signaling pathway and phytohormone signaling [19,20]. The rice *DBB* gene family contains six members, including *OsDBB1* (LOC_Os09g35880), *OsDBB3a* (LOC_Os06g05890), *OsDBB3b* (LOC_Os05g11510), *OsDBB3c* (LOC_Os01g10580), *OsSTO* (LOC_Os04g41560) and *OsSTH* (LOC_Os02g39360) [9].

Poplar is a model plant in woody plant research which grows rapidly and whose genes are easily studied. The *Populus trichocarpa* genome provides the possibility for gene identification and function and evolution analyses. The *DBB* gene family has been identified and described in some plants, such as maize [19,20], rice [9], *Arabidopsis* [5,21], bamboo [22], and *Saccharum* [23], but no systematic research or analysis of the *DBB* gene family in Populus have been conducted. In this study, 12 putative *PtrDBB* genes were identified and classified into four groups, taking *A. thaliana*, *O. sativa*, *Z. mays*, *P. patens*, *S. moellendorffii*, *P. abies* and *A. trichopoda DBB*s and the conserved domains as references [24]. Synthesis and comprehensive bioinformatics analyses were performed to study their phylogenetic relationship, gene and protein structures, domain composition, and chromosome location promoter *cis* elements. Moreover, the expression patterns of 12 *PtrDBB* genes under abiotic stress (light/dark and drought) and phytohormone (ABA and MeJA) treatments were inspected in poplar. On the basis of the information obtained from this study, we provide a precious resource for comprehensive research on the functionality of the poplar *DBB* gene family. 

## 2. Results

### 2.1. Identification of PtrDBB Genes in Populus

To perform the identification of putative *PtrDBB* genes in the Populus genome, an extensive search and the alignment of Arabidopsis DBB proteins sequences were conducted by obtaining these sequences from the *P. trichocarpa* genomic database. After removing the redundant sequences, 12 *PtrDBB* genes (*PtrDBB1*–*PtrDBB12*) were identified, all of which were used to confirm the existence of the two conserved B-box domains without a CCT domain through genome-wide analysis. The domain structures of 12 poplar DBB proteins generated using NCBI and Batch CD-Search are shown in Appendix A. To provide a deeper understanding of the similarity within the poplar DBB gene family, we used MEGA-X 11.0.13 and JalView 2.11.3.0 to perform the multiple alignment of the domain sequences of 12 putative *PtrDBBs* and compared the conservative sequences of B-box1 and B-box2. As shown in Figure 1, the lengths of B-box1 and B-box2 were 38 aa and 36–38 aa, respectively, and the interval length of the two B-box domains was 15 aa. Moreover, the lengths of Box1 and Box2, as well as some differences between the groups, were generally conservative. In addition, details of the *PtrDBB* genes are exhibited in Table 1, including their gene ID, chromosome location, coding sequence length, protein length, physicochemical parameters, molecular weight (MW) and exon numbers. The length of the coding sequences varied from 555 bp (*PtrDBB5*) to 936 bp (*PtrDBB6*). The poplar DBB genes encoded amino acid sequences ranged from 185 to 311 aa, and the predicted molecular weight (MW) varied from 20.51 to 34.34 kDa. In addition, the theoretical isoelectric points ranged from 4.77 to 7.05. 

### 2.2. Analysis of Phylogenetic Relationship and Gene Structures of the DBB Genes

To explore the phylogenetic relationship of the poplar DBB family, a neighbor-joining phylogenetic tree was generated with MEGA-X by aligning 12 PtrDBB protein sequences with 14, 6, 12, 6, 4, 2 and 3 protein sequences from *Arabidopsis thaliana* (dicotyledonous subfamily), *Oryza sativa* (monocotyledons subfamily), *Zea mays* (monocotyledons subfamily), *Physcomitrella patens*, *Selaginella moellendorffii*, *Picea abies* and *Amborella trichopoda* (dicotyledonous subfamily), respectively. Detailed information of the *DBB* genes from *A. thaliana*, *O. sativa*, *Z. mays*, *P. patens*, *S. moellendorffii*, *P. abies* and *A. trichopoda* are listed in Appendix A. Using this phylogenetic tree, the DBB gene family was classed into six subfamilies based on the evolutionary relationships and motif analysis of DBB proteins, and the PtrDBBs were also divided into four groups (G1, G3, G5 and G6) but not G2 and G4. The numbers of PtrDBB members in different groups varied; G1, G3, G5 and G6 contained four, two, two and four genes, respectively (Figure 2).

To gain further understanding of the phylogenetic relationship of *PtrDBB* genes, a separate phylogenetic tree was constructed only using the PtrDBB protein sequences, and the PtrDBB proteins were divided into four subfamilies, which is in accordance with the phylogenetic tree of *A. thaliana*, *O. sativa*, *Z. mays*, *P. patens*, *S. moellendorffii*, *P. abies* and *A. trichopoda*. To gain further information of the structural features of poplar DBB family members, the analysis of exon/intron organizations was performed by matching the *PtrDBB* genes’ genomic DNA sequences with their cDNA sequences. Figure 3 displays the exon/intron predictions of all 12 *PtrDBB* genes. The number of exons in the four distant groups varied from two to six. *PtrDBB10* had the highest quantities of exons (6), and *PtrDBB3* and *PtrDBB9* only contained two exons. However, highly similar exon/intron structures were shared in the majority of the *PtrDBB* genes that were assembled in the same group. For instance, two *PtrDBBs* contained three exons in G2 and G3, respectively, and the members within G4 had four or five exons. Overall, the phylogenetic relationship and conservated gene structure results provide a reliable basis for the classification of poplar DBB family members. 

### 2.3. Analysis of Conservative Motif and Homology Modeling

Two putative motifs of the 12 PtrDBB proteins within each group whose lengths varied from 13 to 50 amino acids were identified using MEME 5.5.5 software, and the length of the conserved sequences and details of each motif are shown in Appendix A. Comparing the MEME and conserved domain structure analysis data (Appendix A), two putative motifs were functionally annotated, which were defined as motif 1 and motif 2 that were close to N- for B-box1, motif 3 that was close to C- for B-box2, and motif 1, motif 2 and motif 3 that, together, formed the double B-box domain. However, no functional annotation was performed for the remaining five hypothetical motifs (Figure 4). The spatial distributions and motif components were varied among the different groups, but they were highly conserved within each group, which inferred that the functions of these proteins might be similar. For example, the 12 PtrDBB proteins all contained the B-box1 and B-box2 domains (motif 1, motif 2 and motif 3). The members of each subfamily not only contained the conserved double B-box domains but also had some specific motifs that may represent their diverse functions in plant development and responses to abiotic stress (Appendix A). Motif 8 (BBOX) was only presented in G3, and motif 4 (Bbox_SF superfamily) existed only in G4. Moreover, eight segmental duplications (PtrDBB1/PtrDBB11, PtrDBB1/PtrDBB3, PtrDBB2/PtrDBB6, PtrDBB5/PtrDBB7, PtrDBB8/PtrDBB12, PtrDBB9/PtrDBB11, PtrDBB9/PtrDBB12 and PtrDBB11/PtrDBB12) exhibited similar or identical motif composition structures.

To obtain a better insight into the tertiary structures of the PtrDBB proteins, the protein sequences were aligned using an HMM-HMM search in intensive mode to perform the homology modeling by Phyre2 [25]. The result showed that all of the 12 PtrDBB proteins could be confidently modeled and had 100% of their predicted lengths modeled, and 100% of their predicted lengths were modeled with >99% confidence (Figure 5).

### 2.4. Chromosomal Location and Gene Duplications of PtrDBBs

Among the eight poplar chromosome scaffolds (Chr1, Chr2, Chr4, Chr5, Chr7, Chr9, Chr11 and Chr17) (Figure 6), chromosomes 4, 5, 7 and 9 had two *PtrDBB* genes, and chromosomes 1, 2, 11, and 17 possessed only one *PtrDBB* gene, whereas only one gene was located on the longest chromosome 1, and two genes were present on the shortest chromosome 9, which implied that the length of chromosome was not proportional to the number of genes. 

The analysis of collinearity was conducted to elucidate the duplication events of *PtrDBB* homologous sequences using BLAST, taking the monocotyledons (*Z. mays*) and dicotyledons (*A. thaliana*) as controls. As a result, 31 and 19 pairs of *PtrDBB* genes were identified as segmental duplications between the poplar and maize genomes, and then the poplar and *A. thaliana* genomes, respectively (Figure 7B). Furthermore, highly conserved collinearity was found among the *PtrDBB* gene regions between *P. trichocarpa* and *Z. mays*, and then *P. trichocarpa* and *A. thaliana*, especially between Ptr4 and Zm5, Ptr11 and Zm1/Zm5, Ptr12 and Zm4, and then Ptr15 and Zm4, all with six synteny genes, Ptr4 and At4/At5, Ptr10/Ptr15/Ptr16 and At3/At4/At5, Ptr11 and At4/At5, Ptr12 and At3/At5, and then Ptr18/Ptr19 and At4. However, 38 duplicated pairs of *PtrDBB* genes were identified as segmental duplication gene pairs in a synteny map (Figure 7A and Table 2), which implied that the amplification of the poplar DBB gene family depends on segmental replication events.

### 2.5. Evolution Profiles and Divergence of DBB Genes

The Ka/Ks ratios and Ks values of the 38 *PtrDBB* gene pairs were calculated and used to evaluate the divergence times and selective pressures of *DBBs* between *P. trichocarpa* and *Z. mays*, and then *P. trichocarpa* and *A. thaliana*. However, the Ka/Ks ratios of the 38 paralogous pairs (Ptr-Ptr) were the lowest at 0.103232 (Ptr2-Ptr9), followed by 0.112035 (Ptr11-Ptr12), and they were the highest at 0.378997 (Ptr11-Ptr7). In addition, the frequency distribution of the calculated average Ks values of paralogous pairs (Ptr-Ptr) was about 0.2, which suggested that the *PtrDBB* genes underwent a massive duplication event of approximately 15 million years ago (MYA). Compared with the previous research that suggested the genome-wide duplication of *P. trichocarpa* occurred 7–12 MYA, the large-scale duplication of *PtrDBBs* happened earlier [4]. However, the frequency distributions of Ks values for the orthologous pairs from the *P. trichocarpa* and *Z. mays*, and then *P. trichocarpa* and *A. thaliana* genomes averaged ~2.43 and ~2.55 (Figure 8 and Table 3), which implied that the divergence times of the *DBBs* were 14 and 15 MYA, respectively. Compared with previously reported results, the divergence time between Populus and maize was 142–163, which inferred that the *PtrDBB* genes experienced gene evolution before the separation of these two progenitor species. The Ka/Ks ratios peaked in the poplar genome (Ptr-Ptr), and for the Ptr-Zm and Ptr-At genomes, they were distributed between 0.1 and 0.38 (Table 2) and 0.06 and 0.58 (Table 3), which suggested that the *DBB* genes might have gone through highly positive purifying selection.

### 2.6. Expression Patterns of DBB Genes in Populus trichocarpa

A hierarchical clustering heatmap was produced using the relative expression of *PtrDBB* genes in the different tissues of *P. trichocarpa* according to the published transcriptome data (Figure 9; Appendix A). This discussion showed that all 12 *PtrDBB* genes in the various tissues at development stages of *P. trichocarpa* were different. In view of the expression characteristics and hierarchical clustering of 14 poplar tissues, 12 *PtrDBB* genes were classified into five clusters (C1–C5). The two genes (*PtrDBB11* and *PtrDBB12*) clustered in C1 were highly expressed in all the tissues except for Xylem1 and Phloem3. Seven genes (*PtrDBB2*, *PtrDBB3*, *PtrDBB4*, *PtrDBB5*, *PtrDBB6*, *PtrDBB11* and *PtrDBB12*) grouped in C5 and C1 exhibited high expression levels in G43h, which showed that these genes are involved in the formation of G43h during poplar growth and development. Additionally, some genes exhibited similar expression patterns in different tissues; for instance, many genes (except for *PtrDBB11* and *PtrDBB12*) exhibited high expression levels in Xylem1, and four genes (*PtrDBB2*, *PtrDBB4*, *PtrDBB5* and *PtrDBB6)* were mainly expressed in G43h and Xylem1. Furthermore, several *PtrDBB* genes showed tissue-specific expression patterns, such as *PtrDBB8*, *PtrDBB9* and *PtrDBB10*, which were only expressed specifically in Xylem1, while *PtrDBB1* and *PtrDBB7* were ubiquitously highly expressed in Xylem1 and Phloem3. Taken together, these results indicated that 12 *PtrDBBs* presented a variety of expression patterns in the poplar tissues at different developmental stages, which encourages future explorations of their functions and characteristics.

### 2.7. Cis-Regulatory Elements Analysis of PtrDBBs

The 2000 bp genomic sequence upstream from the *PtrDBBs* 5′-UTRs showed 41 associated *cis*-regulatory elements (CREs) in the promoter regions using PlantCARE (Figure 10, Appendix A). Among them, six CREs corresponded to light stress. These included two phytohormone-responsive elements, such as abscisic acid- (ABA) (ABRE), MeJA acid- (CGTCA-motif), SA- (TCA), IAA- (TGA and AuxRR-core) and GA-responsive elements (P-box). Detailed information of the genomic coordinates of the analyzed regions in *PtrDBB* genes is given in Appendix A. Additionally, these also included various kinds of abiotic-stress-responsive elements, including a drought-related regulatory element (MBS), a low-temperature response element (LTR) defense and stress responsiveness (TC-rich motif). It was noteworthy that each *PtrDBB* gene had significantly different potential CREs, especially corresponding to light and ABA responsiveness (Figure 10; Appendix A). For example, *PtrDBB1* and *PtrDBB13* contained two low-temperature-responsive CREs, respectively, *PtrDBB6*, *PtrDBB7* and *PtrDBB8* contained two drought-responsive CREs, respectively. *PtrDBB4*, *PtrDBB9*, *PtrDBB10*, *PtrDBB11* and *PtrDBB12* contained two to four ABA-responsive CREs, whereas all the 12 *PtrDBBs* had one or more light stress elements, which suggested that most of the *PtrDBBs* genes took part in light regulation. 

### 2.8. Expression Patterns of PtrDBB Genes under Abiotic Stress and Phytohormone Treatments

According to the analysis of CREs in the promoter regions, some *PtrDBBs* might be involved in potentially responding to different abiotic stresses and phytohormone treatments. Therefore, the expression patterns of 12 *PtrDBBs* under drought stress, light/dark, ABA and MeJA treatments were measured (Figure 11; Appendix A).

Under the light/dark treatments, except for the up-regulated expression levels of *PtrDBB3*, *PtrDBB5* and *PtrDBB9*, the other nine genes exhibited a change in expression level of less than five-fold in comparison with that at 0 h. Among them, *PtrDBB9* showed an expression level that was more than 10-fold higher during the light treatment than that during the dark treatment.

During drought stress, five *PtrDBB* genes (*PtrDBB3, PtrDBB5, PtrDBB6, PtrDBB7* and *PtrDBB9*) exhibited a gradually increased expression level until 12 h and then a reduction. Furthermore, these five *PtrDBB* genes were the most highly expressed (>10-fold that at 0 h) after 12 h of treatment. However, the expression levels of seven genes (*PtrDBB1, PtrDBB2, PtrDBB4, PtrDBB8, PtrDBB10, PtrDBB11* and *PtrDBB12*) changed only slightly (<5 fold that at 0 h) during the 24 h time course.

When the poplar samples were submitted to the ABA treatment, six genes (*PtrDBB4*, *PtrDBB5*, *PtrDBB9*, *PtrDBB10*, *PtrDBB11* and *PtrDBB12*) were distinctly up-regulated. For example, five genes (*PtrDBB5*, *PtrDBB9*, *PtrDBB10*, *PtrDBB11* and *PtrDBB12*) were gradually up-regulated during the early time points, peaked at 12 h, and then decreased over time. *PtrDBB4* was the most highly expressed at 6 h with a gradual increase during the early time points and a significant decrease during the subsequent treatments. Moreover, *PtrDBB9, PtrDBB10* and *PtrDBB11* showed an expression level that was more than 10-fold higher at 12 h than that at 0 h. However, the expression levels of six genes (*PtrDBB1, PtrDBB2*, *PtrDBB3, PtrDBB6, PtrDBB7* and *PtrDBB8*) changed only slightly (<5 fold than that at 0 h) during the 24 h time course.

In the MeJA treatment, four genes (*PtrDBB6, PtrDBB7, PtrDBB9* and *PtrDBB11*) were distinctly up-regulated. For example, the expression level of two genes (*PtrDBB9* and *PtrDBB11*) increased early, peaked at 12 h, and then decreased with time. The expression of one gene, *PtrDBB6*, was up-regulated significantly, reaching a value that was approximately 13-fold higher at 6 h, while the expression of eight genes (*PtrDBB1, PtrDBB2*, *PtrDBB3*, *PtrDBB4*, *PtrDBB5*, *PtrDBB8, PtrDBB10* and *PtrDBB12*) presented insignificant changes (a change in expression levels of less than five-fold at all times). 

Overall, *PtrDBB5* and *PtrDBB9* were the only two genes which were significantly up-regulated in response to all the abiotic stress and different phytohormone treatments. 

## 3. Discussion

### 3.1. PtrDBBs in P. trichocarpa

The DBB transcription factor family, which is a subfamily of the B-box family, plays an essential role in regulating the circadian rhythm and early light morphogenesis. So far, the characteristic and functions of *DBB* genes have been determined in some plants, such as *Arabidopsis* [5,13], rice [9], maize [20], pepper [26,27], cotton [28], tomato [29], apple [30], berry [31] and so on [32]. Six, eight, ten and nine more poplar DBB subfamily members were found compared to those in *Physcomitrella patens* (six), *Selaginella moellendorffii* (four), *Picea abies* (two) and *Amborella trichopoda* (three), respectively. Additionally, the motifs and gene structures in the same subfamily were varied among different branches, but they were highly similar in the same phylogenetic branch. For example, in subfamily I, PtrDBB2 and PtrDBB6 had the same motifs (motifs 1, 2, 3, 4, 5, 6 and 7), and PtrDBB9 and PtrDBB3 contained all the same motif structures (motif 1, 2, 3 and 5). It was determined that PtrDBBs in the same phylogenetic branch have a similar motif structure, leading to a consistent evolutionary pattern of PtrDBB transcription factors. Moreover, the intron/exon structure of PtrDBBs in the same subfamily had some differences, but this was highly similar within the same phylogenetic branch. Thus, the conservation and diversity of motif and intron/exon structures are of great significance for studying the evolution of gene families.

### 3.2. Evolutionary Patterns of DBBs in P. trichocarpa

In view of the phylogenetic relationship analysis data, the poplar DBB proteins were closely related to Arabidopsis DBB proteins, which are also a dicotyledon, and among them, PtrDBB1/PtrDBB11 and AtDBB3, PtrDBB12/PtrDBB6 and AtSTH2, PtrDBB3/PtrDBB9 and AtDBB6, and PtrDBB5/PtrDBB7 and AtDBB1a/AtDBB1b showed more similarity. Furthermore, this genetic relationship is followed by *O. sativa* and *Z. mays*, and PtrDBB4/PtrDBB10 was very similar to OsDBB1 in *O. sativa* and ZmDBB10 in *Z. mays*. In addition, the poplar DBB family was very different from *Selaginella moellendorffii*, *Picea abies* and *Amborella trichopoda*.

Gene duplication is a crucial method of gene family amplification in the genomic evolution of plants [33]; moreover, this helps organisms adapt to various environments during their development [6]. In the poplar DBB gene family, 20 segmental gene pairs and no tandem gene pairs were identified, suggesting that gene segmental duplication was the major driving force for the expansion of the *DBB* gene family in *P. trichocarpa*. The chromosomal localization of the *PtrDBB* gene suggests that the distribution of the DBB gene in the poplar genome may be the result of genome replication events [34]. Six pairs of potential duplicate *DBB* genes appeared in different poplar chromosomes. In addition, the collinearity analysis of Populus and maize genome sequences identified 21 PtrBBX-ZmBBX gene pairs and showed that there was a significant collinearity between *P. trichocarpa* and monocots *Z. mays*, which was consistent with the evolutionary relationship between the dicotyledons and monocotyledons. 

### 3.3. Potential Roles of PtrDBBs in P. trichocarpa

*DBBs* participate in plant growth and development, such as in seedling photomorphogenesis, flowering time, phytochromes, pigmentation and cotyledon development in *Arabidopsis* [5,13], rice [9], maize [20], pepper [26,27], cotton [28] and tomato [29]. In this study, the expression levels of 12 *PtrDBBs* were examined in 14 tissues at different development stages on the basis of previously reported transcriptome data (Appendix A). In view of the results, the *PtrDBB* genes in these different tissues might take part in the control of plant growth and development, and some *PtrDBBs* might have particular functions in specific tissues and at different developmental stages. For example, all of the *PtrDBBs* (except for *PtrDBB11* and *PtrDBB12*) showed relatively high expression levels in xylem1, implying their potential role in xylem1 development. *PtrDBB11*, an ortholog of *AtDBB3,* was highly expressed in most of the tissues except for phloem3 and xylem1. Likewise, *PtrDBB12* also showed a similar expression pattern in these tissues, indicating their essential functions in *P. trichocarpa*. Moreover, *PtrDBB2*, *PtrDBB3* and *PtrDBB6* were the most homologous to AtSTH2, and the expression levels in G43h and xylem1 were significantly higher than those in the other tissues. *AtSTH2* is a regulator of light signal transduction, indicating that its homologous genes *PtrDBB2*, *PtrDBB3* and *PtrDBB6* might be involved in early photomorphogenesis [35]. Other researchers have found that AtSTH2 is able to communicate with HY5 in both yeast and plants, indicating that some DBB proteins, along with HY5 (LONG HYPOOTYL 5), are involved in the complicated early photomorphogenesis of higher plants [14]. In *Arabidopsis*, HY5 positively regulates light signal transduction but negatively regulates photomorphogenesis without light [13]. Therefore, *PtrDBB2*, *PtrDBB3* and *PtrDBB6* might have similar light signal transduction functions in *P. trichocarpa*.

### 3.4. Stress and Phytohormones Induced Expression of PtrDBBs in P. trichocarpa

Many abiotic stresses, such as cold and high temperatures, drought, waterlogging, salinity, metals and nutritional deficiencies can cause a variety of stress response mechanisms, afterwards activating the correlation genes required for stress tolerance [36]. A large number of stress response CREs (41) were present in the promoter regions of the 12 *PtrDBB* genes, indicating their potential roles in stress response. For example, all the 12 genes contained the light-responsive CREs in their promoters, ranging from one to seven. Li et al. [20] has found that *ZmDBB5*, *ZmDBB9* and *ZmDBB10* are up-regulated when poplar is treated with a light/dark treatment. Similar expression patterns of *PtrDBB3* and *PtrDBB9* (homologous of *ZmDBB5* and *ZmDBB9*) and *PtrDBB5* (homologous of *ZmDBB10*) were identified in this study, implying their vital functions in response to a light/dark treatment.

Phytohormones act as important regulators in normal plant growth and development, and auxin, gibberellin, cytokinin, abscisic acid and ethylene are the major phytohormones. The gene family members in *Arabidopsis* and other plant species can respond to different phytohormones [37]. The previous studies showed that DBB1a acted as a negative regulator of circadian rhythm and light signals and participates in the gibberellin homeostasis [13]. In addition, six *DBB* genes in maize exhibited differential expression patterns under various phytohormone treatments, indicating that these *ZmDBB* genes may take part in different phytohormone signaling pathways [20]. In this study, 12 *PtrDBB* genes presented differential expression patterns when challenged with ABA and MeJA treatments, and the majority of them contained CREs that respond to phytohormones in their promoter regions. The expression levels of three genes (*PtrDBB5*, *PtrDBB9* and *PtrDBB11*) were significantly increased under the ABA and MeJA treatments, and all of them harbored ABRE and CGTCA in their promoter regions. Similarly, *PtrDBB3*, *PtrDBB5* and *PtrDBB9* were induced by the light/dark treatment and drought stress, and they all contained 2–10 light-responsive CREs in their promoter regions. In addition, two genes (*PtrDBB5 and PtrDBB9*) were discovered to respond to the light, drought stress, ABA and MeJA treatments, indicating their potential roles in a cross connection between the phytohormones and light signals. Therefore, it has been shown that these *PtrDBB* genes may act as transcriptional regulators to participate in phytohormone signaling pathways and to regulate the plants’ responses to various abiotic stresses.

## 4. Materials and Methods

### 4.1. Plant Material and Growth Conditions

One-hundred-day-old poplar 84 K (*Populus alba* × *Populus glandulosa*) trees grown in a standard greenhouse (light/dark cycle: 16 h/8 h at 25 °C; 85% relative humidity) at the Beijing Forestry University poplar nursery planting base (Beijing, China) were selected as the experimental material. For drought stress treatment, the seedlings were treated with polyethylene glycol (PEG) 6000 (25%). Moreover, the seedlings were treated with MS medium solution (100 mmol/L), 200 µM abscisic acid (ABA), or 200 µM methyl jasmonate (MeJA), respectively [29,38]. The light/dark treatment and daytime treatments were conducted as mentioned by Li et al. [20]. The seedlings were sprayed with Murashige and Skoog (MS) liquid medium supplemented with 200 µM ABA or MeJA. The control seedlings were only sprinkled with MS medium solution.

### 4.2. Identification of PtrDBB Genes in Populus and Phylogenetic Analysis

The genome sequences of *P. trichocarpa* were obtained from the Phytozome12.1 database (https://phytozome.jgi.doe.gov, 1 January 2023). The DBB protein sequences of *Arabidopsis* were acquired from the TAIR database (https://www.arabidopsis.org/, 1 January 2023). In order to identify the putative poplar *DBB* family members, the previously published DBB protein sequences in *Arabidopsis* were used as queries in the BLAST screen against the poplar protein database [39,40]. Additionally, the domains were examined with NCBI (http://blast.ncbi.nlm.nih.gov, 1 January 2023) to inspect the attendance of the B-box1 and B-box2 in all of the predicted poplar *DBB* genes [41]. The molecular weight (MW), isoelectric point (pI) and length of the CDS were estimated using ExPASY (https://web.expasy.org/protparam/, 1 January 2023) [42,43]. Phylogenetic trees were constructed using the MEGA X 11.0.13 (https://www.megasoftware.net/, 1 January 2023) neighbor-joining (NJ) algorithm and bootstrap analysis (1000 values) [44,45]. 

### 4.3. Motif Structure, Gene Structure, Protein Structure, and PtrDBB Gene Promoter Analysis

All of the motifs of the PtrDBB protein sequences were determined using MEME Suite (http://memesuite.org/tools/meme/, 1 January 2023) with the following parameters: an optimum width of 6–50 and a maximum of 10 motifs [33]. The exon/intron structure was analyzed using CDS and genomic information in the GSDS (http://gsds.cbi.pku.edu.cn/, 1 January 2023) [46]. The protein homology models were calculated by comparing the PtrDBB protein sequences with HMM–HMM searches on intensive patterns on the Phyre2 website (http://www.sbg.bio.ic.ac.uk/phyre2/html/page.cgi?id=index, 1 January 2023). The promoter sequence 2.0 kb upstream of the *PtrDBB* genes was submitted to PlantCARE (http://www.dna.affrc.go.jp/PLACE/, 1 January 2023) to identify the putative *cis* elements [47].

### 4.4. Chromosomal Location, Synteny Analysis and Duplication Events

The detailed chromosomal locations of *PtrDBB* genes were retrieved from the Gene Structure Shower of TBtools [48] by importing the genome annotation files and gene IDs and visualizing them using Circos [49]. MapInspect 1.0 software was used to map the physical location of PtrDBBs on the corresponding poplar chromosomes (https://mybiosoftware.com/mapinspect-compare-display-linkage-maps.html, 1 January 2023). Possible genome replication gene pairs of *P. trichocarpa*, *Z. mays* and *A. thaliana*, and duplication events were detected using MCScanX-2019 software (http://chibba.pgml.uga.edu/mcscan2/#tm, 1 January 2023). MCScanX software was also used to perform the collinearity analysis of GLK proteins with BLAST comparing the sequences [50].

### 4.5. PtrDBB Expression Profiles and qRT-PCR Analysis

A heatmap was created using TBtools to exhibit the expression patterns of *PtrDBBs*. RNAseq data of *PtrDBB* genes in 14 tissues were collected according to Rodgers-Melnick et al. [51]. The NCBI Primer-BLAST tool was used to design the primers of the 12 *PtrDBB* genes, which could amplify the 100–200 bp PCR products (Appendix A). Total RNA was isolated from the samples of each tissue using an RNAprep Pure Plant Kit (TransGen Biotech, Beijing, China) according to the user manual. Total RNAs were used for complementary cDNA synthesis using SuperScript III transcriptase (Invitrogen, Carlsbad, CA, USA) in accordance with the manufacturer’s instructions. qRT-PCR analysis was performed on a Bio-Rad CFX96 using the Light Cycler 480 SYBR Green Master Mix (TaKaRa, Dalian, China). The PCR reaction conditions were as follows: 95 °C for 30 s, followed by 40 cycles of 95 °C for 5 s, and 60 °C for 30 s. The quantitative RT-PCR data were analyzed using the 2^−ΔΔCt^ method. The mean expression values and SE values were calculated from the results of three independent experiments.

## 5. Conclusions

In this study, 12 poplar *DBB* genes were identified and divided into four subfamilies based on their gene and motif structures, conserved domains and phylogenetic relationships. In addition, the bioinformatic analysis of chromosome location, collinearity and gene duplication has aided our understanding of the biological functions of poplar *DBB* genes. Several *PtrDBB* genes expressed tissue specificity and diversity at various stages, and some were involved in the abiotic stress and phytohormone treatment responses. The analysis of transcriptomes and qRT-PCR data suggested that *PtrDBBs* might contribute to poplar development by regulating multiple phytohormone signaling pathways. Overall, the genome-wide analysis of *DBBs* is the basis for the potential function analysis of *PtrDBBs* in *P. trichocarpa*, and further research is underway on several *DBBs* to explore their biological functions.

## Figures and Tables

**Figure 1 molecules-29-01823-f001:**
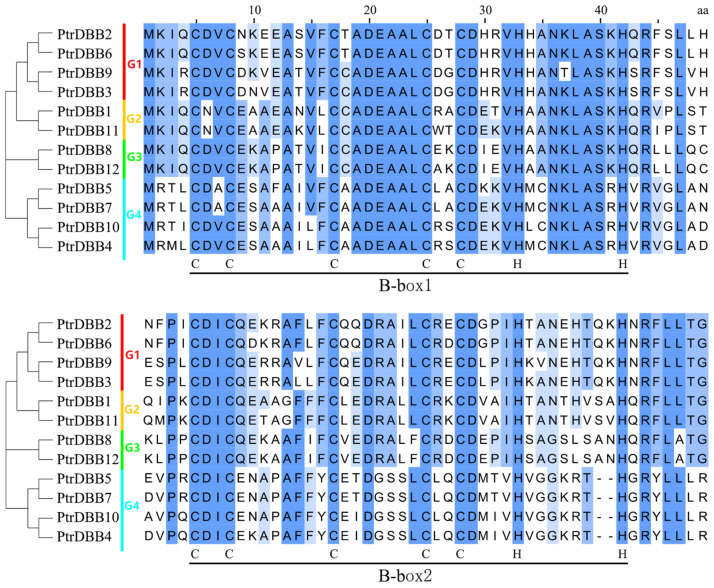
Multiple sequence alignment of 12 PtrDBB conserved domains.

**Figure 2 molecules-29-01823-f002:**
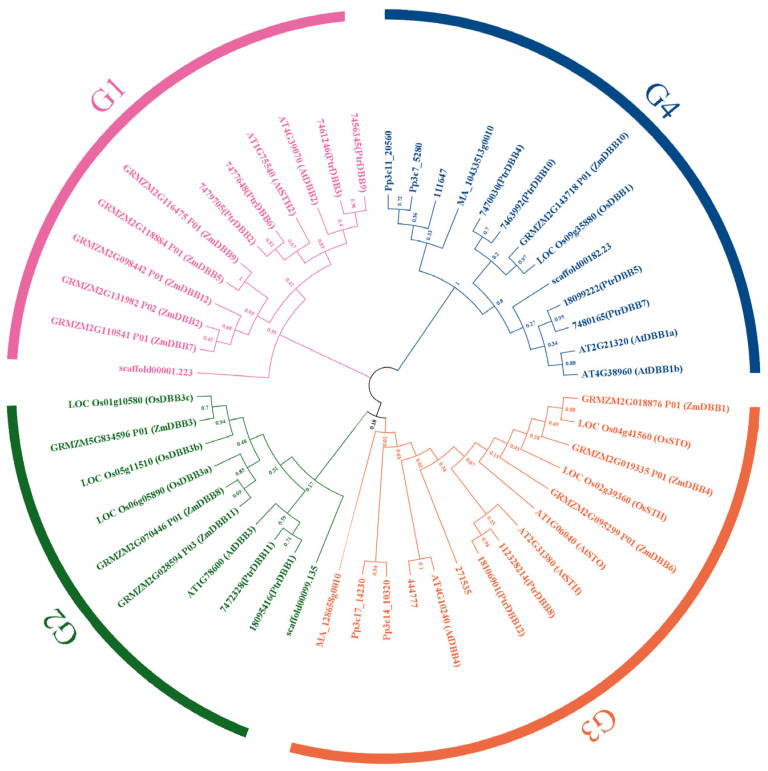
Phylogenetic tree of DBB protein from *Populus trichocarp*, *Arabidopsis thaliana*, *Oryza sativa*, *Zea mays*, *Physcomitrella patens*, *Selaginella moellendorffii*, *Picea abies* and *Amborella trichopoda*.

**Figure 3 molecules-29-01823-f003:**
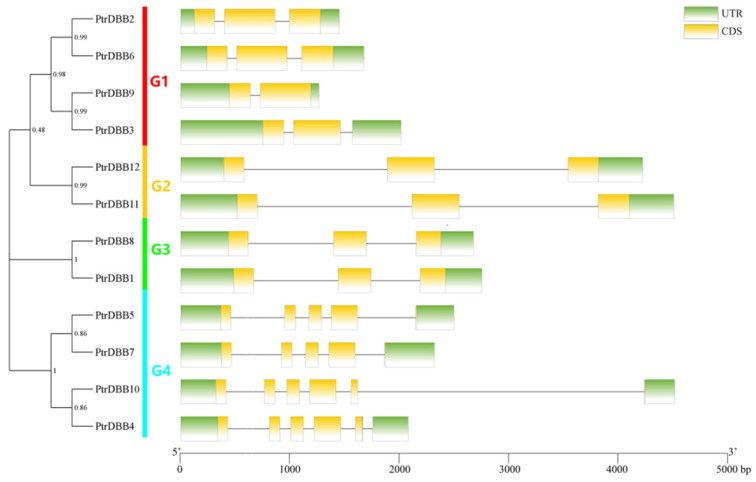
Analysis of phylogenetic relationship and DBB gene structures by MEGA-X in view of the PtrDBB protein sequences in *P. trichocarp*. Left: a neighbor-joining (NJ) phylogenetic tree was constructed. All the *PtrDBB* genes were divided into four clades, and different groups were represented by different colors. Right: the analysis of exon/intron structures of 12 *PtrDBB* genes. The yellow and green rectangles represent exons and introns, respectively. The black lines represents untranslated regions (UTRs).

**Figure 4 molecules-29-01823-f004:**
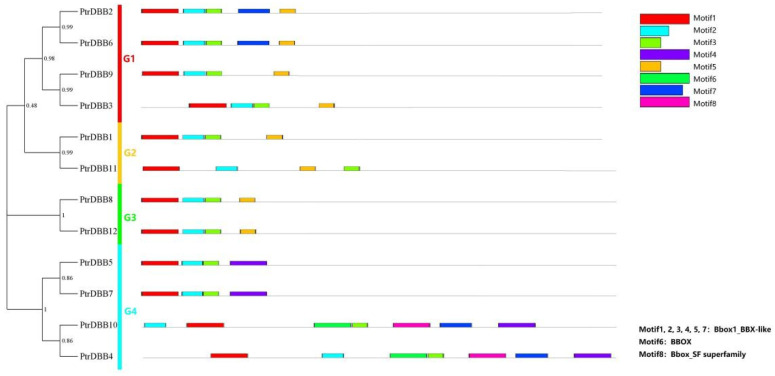
Distribution of conserved motifs for PtrDBB proteins (1–8). The analysis of PtrDBBs conserved motifs was carried out by the MEME. Eight motifs were displayed by boxes of different colors, and the lengths of the motif were represented in proportion. The annotation information of each motif is marked in the bottom right corner.

**Figure 5 molecules-29-01823-f005:**
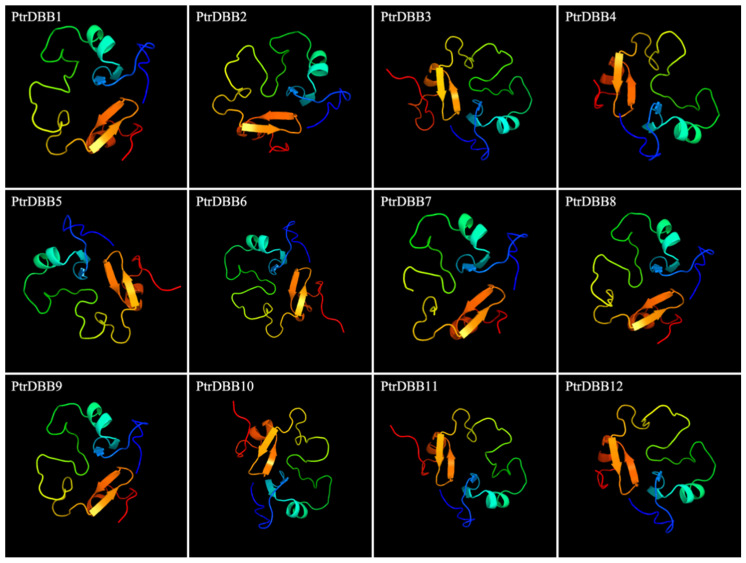
Predicted structures of 12 PtrDBB proteins (>99% confidence).

**Figure 6 molecules-29-01823-f006:**
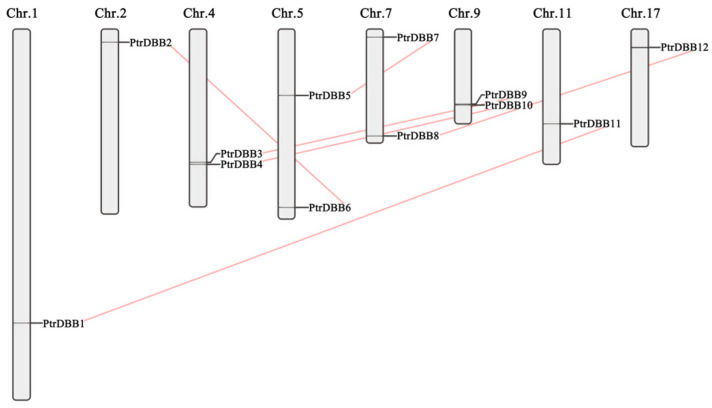
Chromosomal distribution of poplar *DBB* genes. Different colors indicated the grouping of *PtrDBB* genes. Six segmental duplication pairs were connected by orange lines.

**Figure 7 molecules-29-01823-f007:**
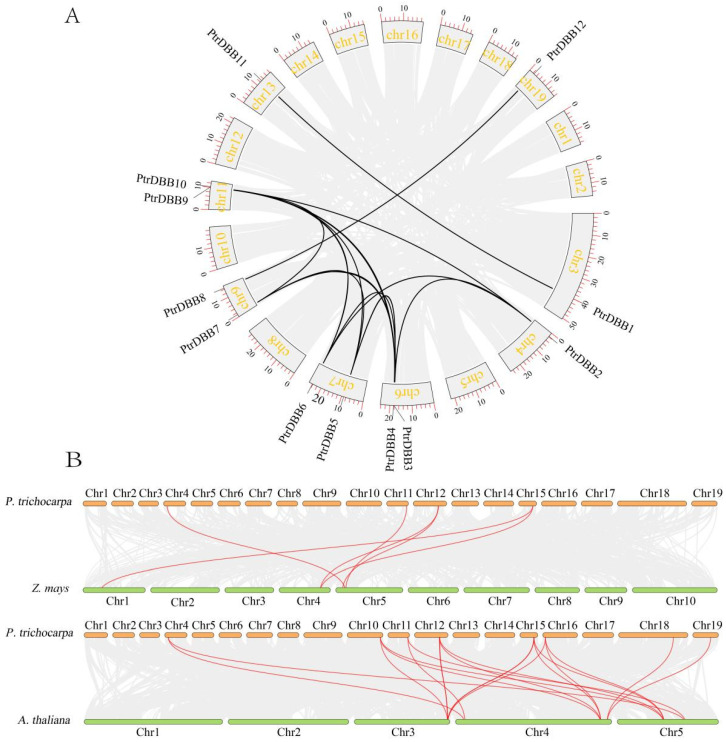
Duplication events of *DBB* genes. (**A**) Synteny of poplar *DBB* genes. (**B**) Synteny of *P. trichocarpa* and *Z. mays*, *P. trichocarpa* and *A. thaliana DBB* gene regions. Segmental duplicated *DBB* gene pairs, and duplicated blocks were linked by red lines and gray lines, respectively.

**Figure 8 molecules-29-01823-f008:**
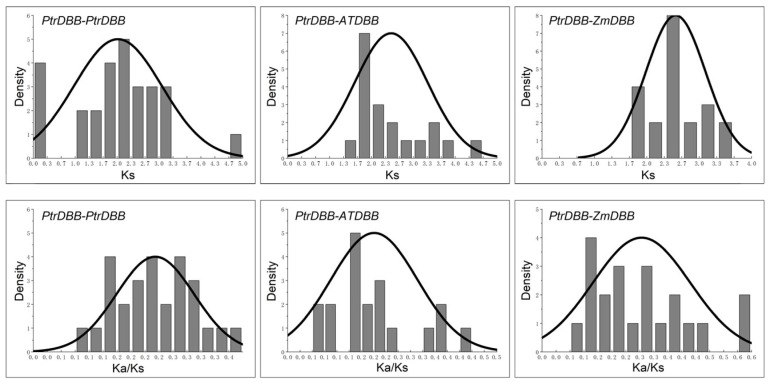
Ks and Ka/Ks value distribution of the *DBB* genes in the genomes of poplar paralogous gene pairs (Ptr-Ptr) and orthologous gene pairs between *P. trichocarpa* and *Z. mays* (Ptr-Zm), *P. trichocarpa* and *A. thaliana* (Ptr-At), viewed from the frequency distribution.

**Figure 9 molecules-29-01823-f009:**
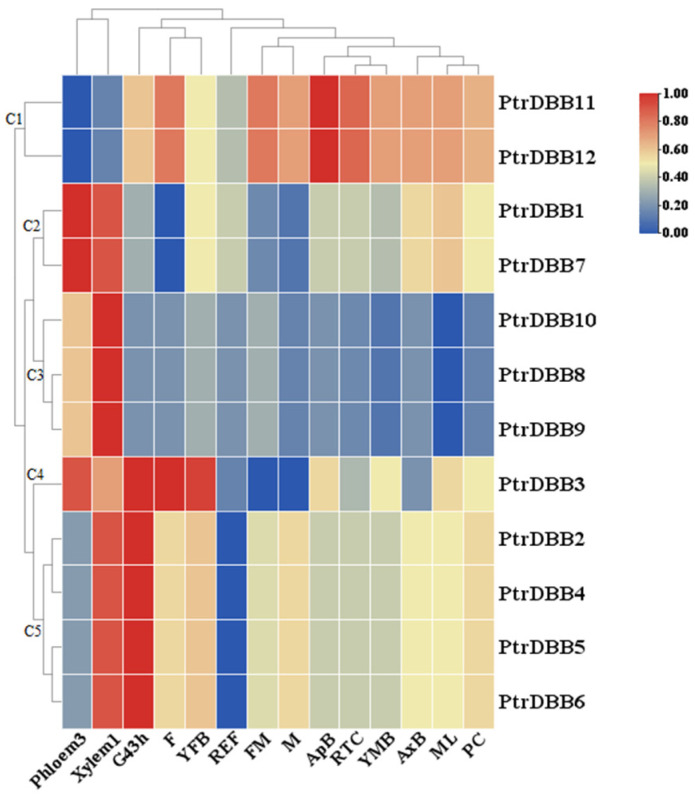
Expression patterns of 12 *PtrDBB* genes in different vegetative tissues and stages of reproductive development. Samples were from 14 tissues, including the following: FM, female catkin, prior to seed release; F, female catkins, post-fertilization; M, male catkins; ML, mature leaf; REF, roots < 0.5 cm diameter from field-grown trees; RTC, roots from plants in tissue culture; G43h, germinated 43 h post-imbibition; ApB, actively growing shoot apex; AxB, axillary bud; YFB, newly initiated female floral buds; YMB, newly initiated male floral buds; Xylem1, developing xylem; Phloem3, developing phloem/cambium; PC, phloem, cortex, and epidermis.

**Figure 10 molecules-29-01823-f010:**
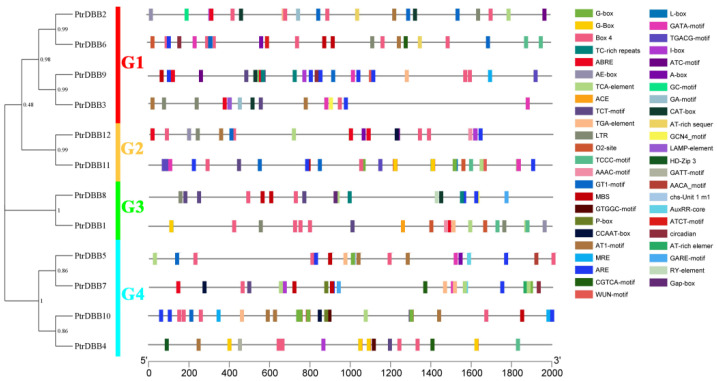
Analysis of *cis*-elements of *PtrDBBs* using the Plantcare database.

**Figure 11 molecules-29-01823-f011:**
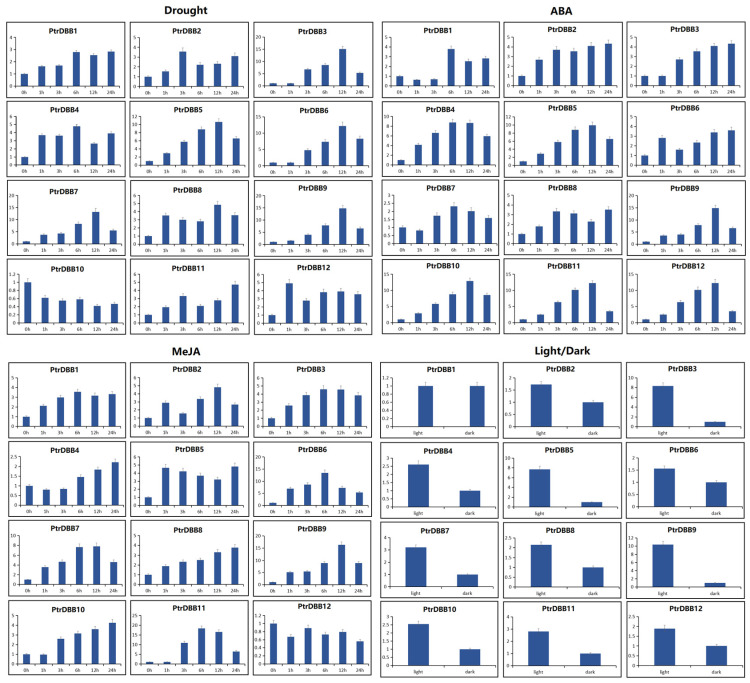
The expression of 12 *PtrDBB* genes under different stresses (light/dark, drought, ABA, and MeJA treatments). Compared with untreated samples (expression levels = 1) after sampling to analyze the relative expression levels. X-axes and Y-axes mean time points after light/dark stress, drought stress, ABA, and MeJA treatments, and data were normalized to reference gene *Ptr18S*, respectively. Three independent biological replicates were conducted.

**Table 1 molecules-29-01823-t001:** Detailed information about 12 *PtrDBB* genes in *P. trichocarpa*.

Gene Name	Gene ID	Location	CDS Length (bp)	Size (aa)	Protein Mw (Da)	pI	Exons
*PtrDBB1*	18095416	1:39,971,068–39,975,286(+)	894	298	31,852.75	5.75	3
*PtrDBB2*	7479705	2:1,813,398–1,814,847(−)	933	311	34,336.37	5.79	3
*PtrDBB3*	7461246	4:18,137,086–18,139,098(+)	621	207	22,859.31	5.72	2
*PtrDBB4*	7470030	4:18,368,982–18,371,060(−)	612	204	22,668.51	5.81	5
*PtrDBB5*	18099222	5:9,032,208–9,034,704(+)	555	185	20,506.21	7.05	5
*PtrDBB6*	7477648	5:24,240,055–24,241,728(+)	936	312	34,288.43	6.20	3
*PtrDBB7*	7480165	7:1,136,453–1,138,769(+)	558	186	20,515.17	6.23	5
*PtrDBB8*	112328214	7:14,516,505–14,519,179(−)	708	236	26,022.39	4.80	3
*PtrDBB9*	7456345	9:10,201,380–10,202,642(+)	654	218	24,021.86	6.39	2
*PtrDBB10*	7463992	9:10,345,821–10,350,331(−)	612	204	22,650.42	6.24	6
*PtrDBB11*	7472328	11:12,884,783–12,889,285(+)	897	299	32,284.26	6.00	3
*PtrDBB12*	18106901	17:2,528,459–2,531,209(+)	717	239	25,989.39	4.77	3

**Table 2 molecules-29-01823-t002:** Ka/Ks of paralogous *PtrDBB* genes pairs (Ptr-Ptr) in *P. trichocarpa*. ^a^ million years ago.

Duplicate Gene Pair	Ka	Ks	Ka/Ks	Purify Selection	Duplication Type	Time (MYA ^a^)
PtrDBB1/PtrDBB9	0.540005	2.643361	0.204287	YES	Segmental	203.3354662
PtrDBB1/PtrDBB10	0.800996	2.706739	0.295927	YES	Segmental	208.2107045
PtrDBB1/PtrDBB11	0.069298	0.231111	0.299846	YES	Segmental	17.77776362
PtrDBB1/PtrDBB3	0.700382	2.136888	0.327758	YES	Segmental	164.3760115
PtrDBB1/PtrDBB6	0.657927	1.818373	0.361822	YES	Segmental	139.874857
PtrDBB2/PtrDBB9	0.280933	2.721383	0.103232	YES	Segmental	209.3371921
PtrDBB2/PtrDBB6	0.052123	0.325995	0.159889	YES	Segmental	25.07652646
PtrDBB2/PtrDBB12	0.569719	2.558617	0.222667	YES	Segmental	196.816677
PtrDBB2/PtrDBB11	0.669162	2.946344	0.227116	YES	Segmental	226.6418438
PtrDBB2/PtrDBB8	0.505474	1.951101	0.259071	YES	Segmental	150.0846888
PtrDBB4/PtrDBB7	0.192639	1.176669	0.163715	YES	Segmental	90.51298954
PtrDBB5/PtrDBB10	0.189565	1.388261	0.136549	YES	Segmental	106.7893365
PtrDBB5/PtrDBB4	0.206347	1.115763	0.184938	YES	Segmental	85.82789923
PtrDBB5/PtrDBB7	0.074573	0.267214	0.279075	YES	Segmental	20.55494785
PtrDBB6/PtrDBB9	0.319537	2.137573	0.149486	YES	Segmental	164.4287226
PtrDBB6/PtrDBB12	0.609082	2.298195	0.265026	YES	Segmental	176.7842262
PtrDBB6/PtrDBB11	0.714865	2.282156	0.313241	YES	Segmental	175.5504411
PtrDBB7/PtrDBB10	0.189482	1.393803	0.135946	YES	Segmental	107.2155891
PtrDBB8/PtrDBB12	0.058761	0.256519	0.22907	YES	Segmental	19.73223654
PtrDBB8/PtrDBB6	0.514803	1.787578	0.287989	YES	Segmental	137.5059917
PtrDBB9/PtrDBB11	0.48228	2.291287	0.210484	YES	Segmental	176.2528183
PtrDBB9/PtrDBB12	0.649449	2.427373	0.267552	YES	Segmental	186.7210054
PtrDBB10/PtrDBB1	0.819611	3.002887	0.272941	YES	Segmental	230.9912805
PtrDBB11/PtrDBB12	0.544453	4.85965	0.112035	YES	Segmental	373.8192222
PtrDBB11/PtrDBB2	0.638452	3.029696	0.210731	YES	Segmental	233.0535028
PtrDBB11/PtrDBB7	0.671339	1.771357	0.378997	YES	Segmental	136.2582048
PtrDBB12/PtrDBB7	0.694002	3.017405	0.23	YES	Segmental	232.1080693

**Table 3 molecules-29-01823-t003:** Ka/Ks of orthologous *GLK* genes pairs (Ptr-At, Ptr-Zm) in *P. trichocarpa* and two species (*A. thaliana* and *Z. mays)*. ^a^ million years ago.

Duplicate Gene Pair	Ka	Ks	Ka/Ks	Purify Selection	Duplication Type	Time (MYA ^a^)
PtrDBB1/AT1G78600	0.273673	1.70025	0.160960704	YES	Segmental	130.7884638
PtrDBB1/AT1G78600	0.273673	1.750689	0.15632326	YES	Segmental	134.6684
PtrDBB2/AT4G38960	0.673689	1.795728	0.375162313	YES	Segmental	138.1329048
PtrDBB2/AT2G31380	0.568092	2.702995	0.210171206	YES	Segmental	207.9227005
PtrDBB4/AT4G38960	0.224073	2.133469	0.105027385	YES	Segmental	164.1130232
PtrDBB4/AT2G21320	0.231998	2.386342	0.097219099	YES	Segmental	183.5647708
PtrDBB4/AT1G78600	0.774086	3.386257	0.228596496	YES	Segmental	260.4812822
PtrDBB5/AT1G78600	0.694743	4.550277	0.152681432	YES	Segmental	350.0212923
PtrDBB6/AT4G38960	0.729036	2.007423	0.363170212	YES	Segmental	154.4171159
PtrDBB6/AT2G31380	0.599408	3.301109	0.181577634	YES	Segmental	253.9314842
PtrDBB7/AT1G78600	0.639815	1.956903	0.326952767	YES	Segmental	150.5310212
PtrDBB9/AT4G39070	0.350527	1.938564	0.180817944	YES	Segmental	149.1203068
PtrDBB10/AT4G38960	0.174062	2.238844	0.077746495	YES	Segmental	172.2187941
PtrDBB11/AT1G78600	0.279831	1.596113	0.175320488	YES	Segmental	122.7779101
PtrDBB11/AT1G78600	0.300215	1.791754	0.167553531	YES	Segmental	137.8272228
PtrDBB11/AT4G39070	0.538742	2.51422	0.214277967	YES	Segmental	193.4015524
PtrDBB11/AT2G24790	0.914811	3.624656	0.252385571	YES	Segmental	278.819697
PtrDBB12/AT4G38960	0.728341	1.671692	0.435690773	YES	Segmental	128.5917088
PtrDBB12/AT2G31380	0.242356	3.907907	0.062016731	YES	Segmental	300.6082636
PtrDBB1/GRMZM2G422644	0.724478	3.282537	0.220707	YES	Segmental	252.5028
PtrDBB1/GRMZM2G143718	0.687263	3.173714	0.216549	YES	Segmental	244.1319
PtrDBB1/GRMZM2G028594	0.398684	3.647221	0.109312	YES	Segmental	280.5554
PtrDBB2/GRMZM2G075562	1.504925	2.630359	0.572137	YES	Segmental	202.3353
PtrDBB2/GRMZM2G028594	1.001651	2.3692	0.422781	YES	Segmental	182.2461
PtrDBB2/GRMZM2G422644	0.716321	2.35668	0.303953	YES	Segmental	181.2831
PtrDBB2/GRMZM2G095299	0.562875	2.550898	0.220658	YES	Segmental	196.2229
PtrDBB4/GRMZM2G098442	0.624664	3.24372	0.192576	YES	Segmental	249.5169
PtrDBB4/GRMZM2G143718	0.2927	1.754125	0.166864	YES	Segmental	134.9327
PtrDBB4/GRMZM2G422644	0.297375	2.10252	0.141437	YES	Segmental	161.7323
PtrDBB6/GRMZM2G075562	1.09016	2.779934	0.392153	YES	Segmental	213.8411
PtrDBB6/GRMZM2G095299	0.675375	2.269925	0.297532	YES	Segmental	174.6096
PtrDBB7/GRMZM2G028594	1.055774	2.892911	0.364952	YES	Segmental	222.5316
PtrDBB8/GRMZM2G028594	1.014354	1.755839	0.577703	YES	Segmental	135.0645
PtrDBB8/GRMZM2G075562	1.107168	2.349007	0.471335	YES	Segmental	180.6929
PtrDBB10/GRMZM2G098442	0.6718	2.342246	0.286819	YES	Segmental	180.1727
PtrDBB10/GRMZM2G422644	0.279454	1.801646	0.15511	YES	Segmental	138.5881
PtrDBB10/GRMZM2G143718	0.255699	1.770641	0.144411	YES	Segmental	136.2031
PtrDBB11/GRMZM2G143718	0.663161	2.435664	0.272271	YES	Segmental	187.3587
PtrDBB11/GRMZM2G028594	0.462405	3.441831	0.134348	YES	Segmental	264.7562
PtrDBB12/GRMZM2G028594	0.840264	2.622381	0.32042	YES	Segmental	201.7216

## Data Availability

Data are contained within the article and Appendix A.

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
