# Peer review of "The DBB Family in Populus trichocarpa: Identification, Characterization, Evolution and Expression Profiles"

_molecules, 2024, doi:10.3390/molecules29081823_

Round 1

Reviewer 1 Report

Comments and Suggestions for Authors

Authors studied the genetic and functional relationship among 12 DBB family genes in Populus trichocarpa. By analyzing the similarity in the coding and upstream regions as well as in the expression patterns under basal and stimulated conditions, they have provided a hypothesis regarding the evolution and functional specification of DBB family genes. This article will contribute to the acceleration of our understanding of PtrDBB genes in P. trichocarpa. 

Author Response

Dear editor:

Thank for the valuable comments and suggestions from molecules. After reading the comments, we came to realize that we indeed had some weak points and unsuitable expressions in our original manuscript. The issues raised were of great help in our revision.

In the past several days, we revised our manuscript after taking into account the comments and suggestions. Below we explain in detail how they were addressed in the revised manuscript.

We checked through and ensure all references are relevant. We are able to address most comments provided by the editors and made changes accordingly in the manuscripts.

        Thank you again for your advice and comments! If you have any queries, please don’t hesitate to contact me at the address below.

Sincerely yours,

Ruihua Wu

Reviewer 2 Report

Comments and Suggestions for Authors

Revisions:

The authors of the manuscript titled " The DBB family in Populus trichocarpa: identification, characterization, evolution and expression profiles" describe their analysis of Populus trichocarpa genome in order to find genes belonging to the double B-box (DBB) family. They study gene structure, protein domains, phylogenesis and expression.

The paper could be interesting but is not well written. Manuscript shows a low attention of authors, starting from author names, and a lot of issues have to be solved. Some methods are not described or not appropriately detailed in M&M and not well thought out.

Moreover, to convince the reader, the authors should accurately revise all the manuscript, also playing attention to the English language.

In particular, here are some examples:

 Line 38: The B-box (BBX) protein belongs to a subfamily of the zinc finger TFs, which participate in transcriptional regulation by binding to other proteins rather than DNA or RNA: this is not true. As definition, transcription factors bind DNA, otherwise they are named coactivators or corepressors. The Zinc-fingers usually are DNA binding domains. Moreover, in vitro studies revealed direct binding of the second B-box of BBX21 to the T/G Box of HY5 promoter, thereby regulating the expression of HY5 and HY5-controlled genes (Proc. Natl. Acad. Sci. USA, 113 (2016), pp. 7655-7660).

Please, use the appropriate nomenclature throughout the paper.

Line 50: The Arabidopsis DBB gene family consists of eight members: please, use official gene name also: BBX18-25.

Line 66: DDB3 and STH2: DBB3 (BBX22) and STH2 (BBX21)…

Methods:

Methods have to be described accurately. Please check the links: i.e. http://memesuite.org/tools/meme should be https://meme-suite.org/meme/. Chek what version of tools you used and describe customization. MEGA 7.0 or MEGA X? A lot of different algorithms and customizations can be chosen in MEGA for both alignment and clusterization.

Line 139 2.5. PtrDBB Expression Profiles and qRT-PCR Analysis: All this section have to be rewritten explaining what RNAseq data were used and how. Describe also how numbers were collected and how heatmap was designed. Moreover, check the Supplementary files.

Also RT-qPCR have to be described in details, starting from treatments.

FIG2: group Numbers should be the same as fig 3 and 4.

FIG4: The annotation information of each motif is marked in the bottom right corner: ?? Moreover, display the B-boxes.

FIG7: P. trichocarpa or P. edulis?

3.6. cis-regulatory elements analysis of PtrDBBs: The regions considered as promoters have not sense.  “The 2000 bp genomic sequence analysis upstream from the translation start site” is not the promoter, since some UTR are very long. Check Additional file5.

Also the Discussion section have to be carefully revised and well thought out. For example, this sentence does not have any sense: (Line 398) “The number of PtrDBB genes is greater than that of these four species, indicating that the genome size of P. trichocarpa is much larger and in accordance with the genome duplication event.”

Comments on the Quality of English Language

English revision is required.

Reviewer 3 Report

Comments and Suggestions for Authors

- Typically, the references used are from two years ago (2022). Therefore, it is advisable to incorporate more recently published articles related to the topic.

- The gene expression analysis by real-time PCR is concise. Please provide additional details for clarity and depth.

Round 2

Reviewer 2 Report

Comments and Suggestions for Authors

Authors answered to some questions, but some issues still need to be resolved.

Question 4:

In the Methods section “A heatmap was created using TBtools to exhibit the expression patterns of PtrDBin” is only reported. Moreover, in the results section authors write only: (lane 328) using the relative expression of PtrDBB genes in the different tissues of P. trichocarpa according to the published transcriptome data (Figure 9; additional file: Table S3).

Therefore: the question remains: what RNAseq data were used and how numbers were collected: At least a reference and an analysis method have to be added. Moreover, the Supplementary file Table 3 was not checked. In the S3 table, some numbers (relative expression???) from 7 tissues (only abbreviations) are indicated:

Table S3.RNA-sequencing data for 12 PtrDBB genes used in this study.

Genes: ApB AxB F FM G43h M ML

Differently, in the heatmap, 14 tissues are included.

Samples were from 14 tissues, including: FM, female catkin, prior to seed release; F, female catkins, post-fertilization; M, male catkins; ML, mature leaf; REF, roots < 0.5 cm diameter from field-grown trees; RTC, roots from plants in tissue culture; G43h, germinated 43 hr post imbibition; ApB, actively growing shoot apex; AxB, axillary bud; YFB, newly initiated female floral buds; YMB, newly initiated male floral buds; Xylem1, developing xylem; Phloem3, developing phloem/cambium; PC, phloem, cortex, and epidermis.

Data have to be coherent.

Please, pay attention RT-qPCR is better than qRT-PCR since the PCR is the quantitative step. RT-qPCR have to be described in details: the amplification protocol have to be added and the method used to verify the amplification (since SYBR green was used) have to be described.

Also, the description of all treatments has to be improved. When, how long, when samples were collected…

Moreover, methods have to be coherent with results:

Methods:

(line 103) The seedlings were treated with MS medium solution (100 mmol/L), polyethylene glycol (PEG) 6000 (25%), 200 µM abscisic acid (ABA), or 200 µM methyl jasmonate (MeJA), respectively (Cao et al., 2017; Chu et al., 2016). The light/dark treatment and daytime treatments were conducted as mentioned by Huang et al. (2012). (Huang et al., 2012). The control seedlings were only sprinkled with MS medium solution.

Results:

the expression patterns of 12 PtrDBBs under drought stress, light/dark, ABA and MeJA treatments were measured (Figure 11; additional file 6: Table S6).

NOTE: The light/dark treatments are not described in Huang et al., 2012. Please, tell the reader if ABA and MeJA are dissolved in MS medium.

Question 8:

3.6. cis-regulatory elements analysis of PtrDBBs: The regions considered was modified, but the results are the same! In any case, in order to explain and better describe the analysis, a table with the genomic coordinates of the analyzed regions have to be added.

Table S5 starts with: ? of cis-acting elements of 12 PtrDBB genes. Please fix the title.

Comments on the Quality of English Language

Moderate editing of English language is required.
